# Into the LAION's Den:
# Investigating Hate in Multimodal Datasets

**Abeba Birhane**[*]
Mozilla Foundation and
School of Computer Science and Statistics
Trinity College Dublin, Ireland

**Vinay Prabhu**[*]
Independent Researcher

**Sang Han**
Independent Researcher

**Vishnu Naresh Boddeti**
Computer Science and Engineering
Michigan State University

**Alexandra Sasha Luccioni**
Hugging Face
Montreal, Canada

## Abstract

'Scale the model, scale the data, scale the compute' is the reigning sentiment in the world of generative AI today. While the impact of model scaling has been extensively studied, we are only beginning to scratch the surface of data scaling and its consequences. This is especially of critical importance in the context of vision-language datasets such as LAION. These datasets are continually growing in size and are built based on large-scale internet dumps such as the Common Crawl, which is known to have numerous drawbacks ranging from quality, legality, and content. The datasets then serve as the backbone for large generative models, contributing to the operationalization and perpetuation of harmful societal and historical biases and stereotypes. In this paper, we investigate the effect of scaling datasets on hateful content through a comparative audit of two datasets: LAION-400M and LAION-2B. Our results show that hate content increased by nearly **12**% with dataset scale, measured both qualitatively and quantitatively using a metric that we term as Hate Content Rate (HCR). We also found that filtering dataset contents based on Not Safe For Work (NSFW) values calculated based on images alone does not exclude all the harmful content in alt-text. Instead, we found that trace amounts of hateful, targeted, and aggressive text remain even when carrying out conservative filtering. We end with a reflection and a discussion of the significance of our results for dataset curation and usage in the AI community. Code and the meta-data assets curated in this paper are publicly available at `https://github.com/vinayprabhu/hate_scaling`. *Content warning: This paper contains examples of hateful text that might be disturbing, distressing, and/or offensive.*

## 1 Introduction

Generative AI models have come to captivate diverse stakeholders, spanning from researchers [7, 42] to media institutions [28, 31] and even large-scale investment firms [18, 34]. The release of numerous generative models introduced generative AI to the general public. The emergence of the Dall·E 2 [57], a text-to-image vision-language model released in April 2022, which purportedly attracted over a million users within the first three months of its launch and was celebrated with claims like: "the first AI technology that has caught fire with regular people" [28]. Subsequently, models such as Stable Diffusion [58] and Midjourney emerged, followed by proprietary projects from large technology companies

---

[*]Equal contribution

37th Conference on Neural Information Processing Systems (NeurIPS 2023) Track on Datasets and Benchmarks.

such as Imagen [59], Parti [76], and BASIC [52], access to which was never given to the general public. While Stable Diffusion and its variants have been trained on open-sourced datasets from the LAION family, little is known about the datasets that are used to train models such as OpenAI's Dall-E [57], Google's Parti [76], and Imagen [59]. Fundamental to this multimodal model development is large-scale vision-language datasets containing image-text pairs, which form the main focal point of this paper.

Broadly speaking, these datasets are of two types: those that are open-source, "freely available," and mainly scraped from the Common Crawl (such as LAION-400M [63] and LAION-5B [62]), and those that are closed datasets, curated internally by big corporate labs (such as Google's ALIGN 1.7B/ALIGN 6.6B, JFT-5B [52], and OpenAI's WebImageText-WIT [54]). Although their products increasingly permeate society (and rarely remain within the bounds of corporate labs), information on training datasets remains outside the reach of independent audits and evaluations, where only some of the models trained on such datasets are accessible to the general public via commercial APIs (such as Dall·E 2 or Runway ML). Canonical trend-setting papers introducing state-of-the-art (SOTA) models such as ALIGN [33], Imagen [59] and Parti [76] systematically conceal critical details on training data (See Appendix D for more). These models are also adopted in various commercial tools and applications such as stock photo generation [41], further accelerating their adoption and usage.

The open-source variants of these datasets are getting bigger and now breaching the billion-samples mark for two reasons: firstly, the unquestioned subservience to the *scale is all you need* mandate handed down from models underlying the field exemplified by works such as [33]. Secondly, there is an increased alliance between dataset creation and venture capital, resulting in the financial backing of dataset creation efforts, which was hitherto missing, i.e., that companies are sponsoring dataset creation efforts. The LAION-5B [62] dataset, for example, was sponsored by Hugging Face, Doodlebot, and Stability.ai, as per their blog post announcement.

The *scale is all you need* mandate, in turn, emerges from reasoning that treats dataset creation, curation, and management in a lackadaisical manner in published literature accompanying datasets, often resting on vaguely-defined 'dataset scaling laws' (see Appendix C for further elaboration). This is further exacerbated by the vague and often un-reproducible empirical results buried in subsections of some of the canonical papers which (barely) describe the closed datasets used for training models such as ALIGN [33], Imagen [59] and Parti [76] (See Appendix D for more). We also note that these high-profile and trend-setting disseminations are increasingly turning into a flag-posting exercise that involves tactfully concealing critical details on how the dataset was curated, where the data came from, and what it contains (See Appendix E for a deeper exploration).

All of this has resulted in a *scrape-first-ask-later* data creation and curation culture generating gargantuan datasets, in turn eliciting several copyright lawsuits [40], en masse fetishization of women's bodies in an emergent synthetic digital culture [48], outright bans of model outputs from art forums [17], and marquee datasets filled with hundreds of millions of duplicates [75]. These dataset drawbacks and limitations, in turn, result in downstream negative impacts, often against marginalized identities, for example, worsening of negative stereotypes and biases [42, 21, 2, 3], discriminatory and harmful representations [66, 67, 61] and disparate performance and treatment based on gender, race, and other dimensions [64, 4, 25].

Hateful speech, which is commonly understood as speech that attacks individuals or groups based on attributes such as race, religion, sexual orientation, or gender identity [44, 74], has become a growing threat in online media [10, 19, 45, 2]. The United Nations Secretary-General Antonio Guterres recently appealed to the global community to "tackle the hate that spreads like wildfire across the internet" [47]. Owing to this growing threat of hate speech, toxicity classification and hate-speech detection have emerged as important challenges in the machine learning community (See [9, 22, 32, 53, 73] for systematic surveys).

Hateful, abusive, racist, aggressive, and targeted speech are overlapping phenomena where each can be characterized along dimensions such as directed, generalized, explicit, and implicit abuse or hate [74]. Often, the targets of hateful speech are minoritized groups. Based on analysis of generated data to improve hate detection [70] highlights that Black people, women, Muslims, and trans people constitute groups that are most often the targets of hate. Hateful, abusive, and aggressive speech is a systemic problem. For example, [2] studied the outputs generated from GPT-3 when the word "Muslim" is included in the prompt. They found that 66 out of the 100 completions were violent. Findings from studies of hate speech and abusive language detection datasets show systemic bias. For example, [12] examined hate speech and abusive language detection datasets and found systematic racial bias in

all datasets. Consequently, their findings show that classifiers trained on these datasets predict tweets written in African-American English as abusive at a substantially higher rate. Auditing vision-language datasets for hateful content is, thus, a critical first step to ameliorate the downstream effects of hateful content from models trained on such data. The extent of hate speech in a large-scale vision-language dataset, such as the ones we study, has yet to be established.

This paper sheds new light on the presence of hateful content and the impact of scale through a comparative audit of two open-sourced vision-language datasets, LAION-400M and LAION-2B-en. More specifically, we examine the impact of dataset size on the presence of *hateful, targeted* and *aggressive* content and what portion of this content is filtered out using existing filtering approaches.

The rest of the paper is organized as follows. Section 2 provides some context to the current AI landscape that treats scale as a solution to many problems and critical scholarship that illustrates shortcomings with such narratives. In Section 3, we detail our audit methodology; in Section 4, we present our findings. Our qualitative and quantitative analysis using HCR show that problematic content increases with dataset size. Our NFSW label analysis also reveals some correlation between 'hateful' and 'targeted' speech and NSFW values. We reflect upon these results and propose several recommendations for a more rigorous, transparent, and equitable dataset creation and curation practice in Section 5. We conclude the paper in Section 6.

## 2    Scaling Datasets: An Overview

Current thinking around scale can be broadly categorized under two differing approaches: that which views scale as a solution to a multitude of concerns such as model performance, bias, and generalization and that which emphasizes numerous concerns that arise with an unwavering commitment to scale and large-scale dataset scraping. We present both below.

**Scale as a solution:** Heralded by emblematic large-scale datasets such as ImageNet [14] and C4 [55] over the past decade, the field of machine learning (ML) has come to treat large scale as a mark of *success*, *progress*, and *improvement* as well as a *solution*. This can be observed in the popularity of the field-wide concept of *scaling laws*, where large scale is often juxtaposed with better model performance [35]. Following analysis of the top 100 most influential ML papers from the past decade published in two of the most prestigious ML conferences (NeurIPS and ICML) [6], for example, found that "scaling up" is one of the top sought out values in ML research. Similar sentiment about scale also prevails about datasets. For instance, model performance, according to Schumann et al. [63], can be improved simply by making datasets bigger.

Scale is furthermore presented as a shortcut that can circumvent various dataset-related problems. These include the presence of problematic content in datasets, resource-intensive dataset curation, and costly annotation processes, where a larger scale is seen as a substitute for quality data. According to Jia et al., for example, "heavy work on data curation and annotation" can be avoided by scaling up image-text datasets [33]. This "scale beats noise" narrative has tactfully re-framed the thoughtful handheld dataset curation process as a costly burden that can be "solved" by larger scale. Scale, subsequently, is introduced as a liberating panacea that not only frees the downstream ML pipeline from the burdens of expensive filtering or post-processing steps but also something that makes up for "noisy" data, as if captioning errors in multimodal datasets of image and alt-text pairs can somehow be "averaged out" through the correct captioning elsewhere in the dataset. Similarly, in the 'data curation' section of FLorenceDataset-900M [77], image-alt-text data scraped from the World Wide Web is presented as *'Diverse'* and *'Noisy free-form'*. Such lines of thinking are not unique to this specific context but form a widespread belief that drives initiatives such as the LAION datasets and permeates the entire field of multimodal pursuit.

**The cost of scale thinking:** The primary motivation behind the LAION-400M undertaking was to produce open-source variants of the opaque Web-Image-Text (WIT) dataset, and the CLIP [54] and DALL.E [57] models. Such data open-sourcing initiatives are important first steps towards accountability and building rigorous and equitable AI, given that open access is a crucial prerequisite for independent audit and evaluation. Nonetheless, numerous concerns arise with web-sourced data. Subsequently, continual rigorous audits and assessments of these datasets are imperative for well-functioning, equitable, and healthy open-sourcing practices.

For instance, Science and Technology Studies (STS) scholars and critical data and AI studies have repeatedly emphasized that "scale thinking" stands in stark opposition to values such as societal equity

and effective systemic change [26, 36]. In fact, unwavering commitment to scalability is instrumental to the realization of central objectives driving big technology corporations, such as profit maximization, market monopoly, and the centralization of power in a handful few, all too often at the expense of prioritization of informed consent, justice, and consideration for societal impacts of models [6, 4].

## 3 Dataset Audit: LAION-400M and LAION-2B-en

The last few years have seen more targeted efforts within the ML community where the value and importance of work on datasets has become apparent. Artifacts such as data sheets proposed to accompany and document ML datasets [23] and the emergence of a robust body of work around dataset auditing, data exploration, and documentation [15, 50, 49, 60] has emphasized the need to consider such work an integral part of model development.

This has been accompanied by scholarly work looking at datasets such as the Common Crawl [43], C4 [16], and LAION-400M [7], which have found a multitude of problematic and harmful content and stressed the importance of data documentation and auditing. With this awareness, there has also been increased attention towards the need to evaluate and audit models trained on these datasets as an important intervention and accountability mechanism [56, 46, 68]. There have also been several efforts to carefully create and curate large-scale datasets to reduce the inclusion of problematic content and emphasize multilingualism (e.g., the Pile [20], the ROOTS corpus [38]) which have since been used and disseminated within the community.

Previous audits of multimodal datasets thus far have investigated the images contained within them, using techniques ranging from image content analysis [65, 7] to image source analysis (by analyzing the URL field), as well cross-modal analyses between images and text (i.e. looking for discordance between an image and its alt-text description). Findings from these analyses indicate that these datasets contain a multitude of issues. For instance, poor data quality is a significant concern. Nearly 30% (700 million image-text pairs) in the LAION-2B-en dataset, for example, are duplicates [75]. This, as addressed in [65] and [75], can manifest as *Digital Forgery*, or exact memorization of training examples present multiple times in training data, which was shown to be possible in recent work by Carlini et al. [8] – a phenomenon that has stark ramifications for the field of image generation at large. Other work has shown the presence of explicit images, pornography, malign stereotypes, and other extremely problematic content in the LAION-400M dataset [7].

### 3.1 Audit methodology

Multimodal datasets can present harm and biases in either modality. The focus of the current work is the alt-text accompanying the images in two LAION datasets. We particularly examine 1) the relationship between the presence of harmful content and scale and 2) whether harmful textual content correlates with the Not Safe For Work (NSFW) image labeling already carried out on the dataset. To this end, we use a SoTA open-source NLP toolkit known as *pysentimiento* [51][2].

We use the 'hate speech' analyzer of the framework, which is trained to detect misogyny and racism. In response to an input sentence, it outputs a $3 \times 1$ vector containing probability scores across the three categories of hateful speech, targeted speech, and aggressive speech. While the 'hateful' score indicates if hate speech is present, 'targeted' indicates whether the target of the text is generic or a specific group of individuals (e.g., women or immigrants), and 'aggressiveness' indicates whether the content also includes aggression or violence.

For example, the following alt-text *'Biden's Spending Will Go To Illegal Immigrants While Tax Hikes Will Destroy American Jobs'* results in *pysentimiento* returning [`hateful: 0.902, targeted: 0.024, aggressive: 0.449`], whereas this one: *'slave punished by angry blond mistress'* results in [`hateful: 0.967, targeted: 0.891, aggressive: 0.919`] (both examples are randomly sampled from alt-text descriptions found in LAION-400M dataset).

---

[2]To support our initial study, we looked for a state-of-the-art open-source multilingual Python toolkit that supported GPU inference and one that was peer-reviewed, well cited, well documented, actively under development with responsive maintainers (See `https://github.com/pysentimiento/pysentimiento/issues/39`) and Pysentimiento satisfied all of the above criteria. The project has 80+ citations and 400+ Github stars. It supports Spanish, Portuguese, and Italian, which constitute approximately 1/6th of the `Laion2B-multi` dataset that is the next candidate for the type of analysis presented here. We outline the limitation of this tool in Section 5.

## 3.2   Experiment design

To evaluate the impact of scaling a dataset from 400 million to 2 billion samples on text quality pertaining to hate speech, targeted speech, and aggressive speech, we perform the following audits: we first sub-sample image rows from both datasets and extract the alt-text descriptions associated with the sampled image rows from the TEXT field. In all of our experiments, since we are limited by the amount of computational resources we have access to, we sub-sample 100,000 rows from each of the 160 constituent parquet files spanning the two datasets (the number of files the dataset is separated into). This yields $N_{samples,400M} = 3.2$ million samples for the LAION-400M dataset and $N_{samples,2B-en} = 12.8$ million samples for the LAION-2B-en dataset.

The extracted alt-text descriptions are then passed through *pysentimiento* to extract the $N_{samples} \times 3$ text-quality score matrices for each of the two datasets with the three columns mapping to probability scores of hateful speech, targeted speech and aggressive speech respectively. We then perform a quality control step where we check if any of the three probability score values associated with an input alt-text description exceeds a specific pre-set threshold score $P_{threshold}$, in which case the input text is deemed to have failed the quality check at that threshold.

We define the metric, Hate Content Rate (HCR) [3]: $\psi_{type}(P_{threshold})$ to be,

$$\psi_{type}(P_{threshold}) = 100 \times \frac{\sum_{i=1}^{N_{samples}} \mathbb{1}(\tilde{p}_{type,i} > P_{threshold})}{N_{samples}} \tag{1}$$

where $\mathbb{1}(\cdot)$ is the indicator function, $\tilde{p}_{type,i}$ is the probability score assigned by the *Pysentimiento* model for the text associated with the $i^{th}$ sample and $type \in \{'\text{hateful}', '\text{targeted}', '\text{aggressive}'\}$.

This captures the ratio of samples (in percent) that resulted in the *pysentimiento* model assigning the associated hate/targeted/aggressive speech probability score to be greater than $P_{threshold}$.

We also introduce the *Any-of-the-three* metric, which maps to the case where the input text fails the quality test if any of $\tilde{p}_{hateful}$, $\tilde{p}_{aggressive}$ or $\tilde{p}_{targeted}$ happens to be greater than $P_{threshold}$. The associated 'Any-of-the-three'-HCR, $\bar{\psi}(P_{threshold})$ would be:

$$\bar{\psi}(P_{threshold}) = 100 \times \frac{\sum_{i=1}^{N_{samples}} \mathbb{1}(\max\{\tilde{p}_{hateful,i}, \tilde{p}_{targeted,i}, \tilde{p}_{aggressive,i}\} > P_{threshold})}{N_{samples}} \tag{2}$$

We use both metrics to compare the statistics associated with the text-quality score matrices to understand the nature of the text that was scooped in when the dataset expanded from 400 million samples to 2 billion samples, as well as to examine the relationship between harmful content detected in images and alt-text.

## 4   Results

In the forthcoming section, we use both HCR and, more specifically, 'Any-of-the-three'-HCR, $\bar{\psi}(P_{threshold} = 0.5)$ as the default metric of comparison to characterize the amount of hateful content in both LAION-400M and LAION-2B-en datasets (Section 4.1). We also carry out a file-wise comparison of specific shards of both datasets (Section 4.2) and analyze the correlation between toxic alt-text and NSFW images in Section 4.3.

### 4.1   Scaling is not benign: comparing LAION 400M and LAION 2B-en

For all the sentiment types (hate, targeted, and aggressive speech) detected by *pysentimiento*, the curve corresponding to the 2B-en dataset lies strictly above the curve for the 400M dataset. This signifies that irrespective of what $P_{threshold}$ is being chosen, the quality failure rate shows that the prevalence of hateful content is higher with the 2B-en dataset than its 400M counterpart. As shown in Figure 1,

---

[3]We use the metric Hate Content Rate (HCR) as a shorthand for not just hateful content but all three categories: hateful, targeted, and aggressive.

as the $P_{threshold}$ is increased, the HCR curves monotonically decrease, indicating that fewer textual samples meet the more stringent constraint placed by a higher $P_{threshold}$. Out of the three sentiments – hateful, targeted, and aggressive – hateful content was the most prevalent in both datasets. The 2B dataset showed an HCR of up to **0.7**, and the 400M dataset showed an HCR of **0.6**, followed by targeted speech, with an HCR up to **0.25/0.2**, and finally aggressive speech, with an HCR of **0.04/0.03**.

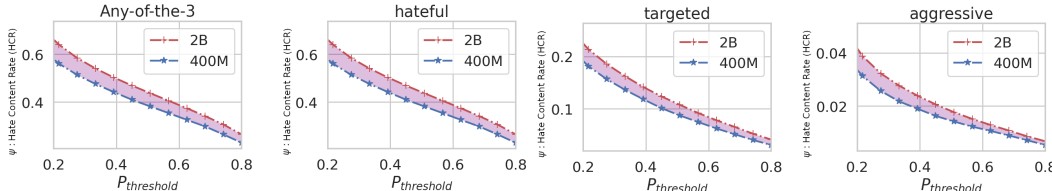

Figure 1: HCR curves for the LAION400M and LAION-2B-en datasets using *pysentimiento* outputs showing that Hate Content Rate increased with dataset size.

Based on the *'Any-of-the-three'* curve (leftmost on Figure 1), we can see that the HCR curve for the 2B-en dataset is above the curves of the 400M dataset. Given that both LAION-400M and LAION-2B-en are extracted from the Common Crawl dataset, we hypothesize that during the race to expand the dataset to 5 billion samples, the dataset scraping module might have sampled from the lower-quality sub-parts of the Common Crawl at a rate worse than that during the LAION-400M dataset creation process. We also note that the CLIP-filtering threshold has been relaxed from 0.3 (for LAION-400M) to 0.28 (for LAION-2B-en), which could be another explanatory factor.

To further investigate the relationship between scale and quality, we conduct a binomial proportion confidence interval analysis to establish lower and upper confidence levels of HCR for both datasets at a given reasonable $P_{threshold}$ of 0.5. For this, we use the Wilson Score interval method with coverage at 0.95 (or $\alpha = 0.05$) that resulted in:

$$
\begin{aligned}
&\bar{\psi}(0.5) \in \left[ \bar{\psi}_{lb,dataset}^{(\alpha=0.05)}(0.5), \bar{\psi}_{ub,dataset}^{(\alpha=0.05)}(0.5) \right] \\
&\bar{\psi}_{400M}(0.5) = 0.298 \in [0.292, 0.304] \\
&\bar{\psi}_{2B-en}(0.5) = 0.344 \in [0.341, 0.347]
\end{aligned}
\tag{3}
$$

where $\bar{\psi}_{lb,dataset}^{(\alpha=0.05)}$ and $\bar{\psi}_{ub,dataset}^{(\alpha=0.05)}$ are the lower-bound and the upper-bound values of the confidence interval at $\alpha = 0.05$.

As seen above in equation 3, the lower-bound HCR for the 2B-en dataset is markedly higher than the upper-bound estimate of HCR for the 400M dataset thus leading to change-of-HCR, $\delta_{CI} = \left( \frac{\bar{\psi}_{lb,2B-en}^{(\alpha=0.05)}(0.5) - \bar{\psi}_{ub,400M}^{(\alpha=0.05)}(0.5)}{\bar{\psi}_{ub,400M}^{(\alpha=0.05)}(0.5)} \right) \times 100$ of 12.26%. Simply put, even under an optimistic setting where we compute the difference between the lower-bound estimate of HCR for the 2B-en dataset and the upper-bound estimate of HCR for the 400M dataset, we still see a **12.26**% normalized increase in HCR.

We have so far established the risks of extending the LAION-400M dataset quality statistics to its bigger counterpart, that is, LAION-2B-en. This, as we hypothesize, may be a consequence of rich non-iid inter-sample correlations emerging from a graph-structured prior for CommonCrawl. This begs the question, given that our dataset was uniformly sampled at the shard/file level, whether the HCR statistics computed at the shard/file level can meaningfully compare with the global dataset-level HCR statistic. We examine this in the sub-section below.

## 4.2 Intra-dataset filewise comparisons

Given that the two datasets, LAION-400M and LAION-2B-en, are split into, respectively, 32 and 128 purportedly uniformly sampled shards, we also examine the validity of the file-level HCR metrics to the global dataset-level metrics. To this end, we use the three 100,000-sized file-level text-quality score matrices obtained from the *pysentimiento* model and compute what fraction of these rows are larger than $P_{threshold}$ of 0.5 for all the three columns. This yields file-level HCRs (in %) for each dataset, with the three columns mapping to hateful speech, targeted speech, and aggressive speech.

We found that the file-wise HCRs were all tightly clustered around the mean levels for the individual datasets. Figure 2 shows the fused swarm-box-violin plot that captures the file-wise HCR metrics

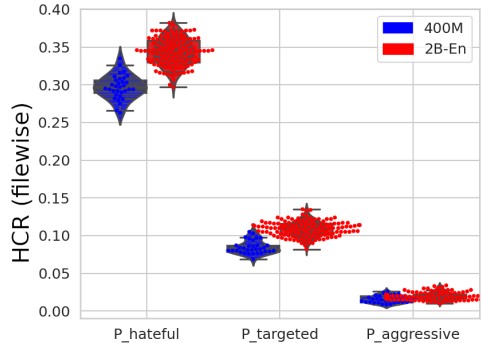

| | hateful | targeted | aggressive |
|---|---|---|---|
| **T** | 14.48 | 13.80 | 4.44 |
| **Degrees of freedom** | 53.32 | 54.91 | 47.96 |
| **P-value** | 2.02e-20 | 8.44e-20 | 2.60e-05 |
| **Cohen-d** | 2.64 | 2.47 | 0.87 |
| **BF-10** | 3.785e+27 | 5.957e+25 | 2131.144 |

Figure 2: Fused swarm-box-violinplot that captures the file-wise HCR metrics for all the 160 (=32+128) parquet files from LAION400M and LAION-2B-en. HCRs for LAION-2B-en (the red swarms) are higher than the 32 file-level HCRs for the LAION400M (the blue swarms) for all three sub-categories – hateful, targeted, and aggressive speech.

Table 1: Results from the two-sample t-test while correcting for unequal variances (using the Welch separate variances T-test). 'BF10' indicates the Bayes Factor of the alternative hypothesis. For all three categories of hateful, targeted, and aggressive speech, the file-wise HCR associated with the 2B-en dataset is higher than the file-wise HCR for the 400M dataset, showing dataset degradation with dataset scaling.

for all the 160 (=32+128) parquet files spanning the two datasets. For example, the 'hateful' related HCR for LAION-400M has a mean value of **0.298**, which increased to **0.344** for LAION-2B-en. All the 32 constituent file-wise HCRs for this dataset fall within **0.26** and **0.33**.

Furthermore, $97\%$ of all the files have their HCRs within two standard deviations of the mean-HCR for the dataset. Similarly, the mean-HCR for the entire LAION-2B-en dataset is 0.344, and the range across all the 128 files is [0.297,0.382], which indicates that $\sim 95\%$ of all the constituent files to have their file-level HCRs to be within two standard deviations of the dataset level mean HCR.

We also observe that the 128 file-level HCRs for LAION-2B-en (the red swarms in Figure 2) are higher than the 32 file-level HCRs for the LAION400M (the blue swarms) for all the three sub-categories of hate speech. To ascertain if this difference is statistically valid, we perform a two-sample t-test while correcting for unequal variances (using the Welch separate variances T-test) and explicitly setting the alternative hypothesis to be greater (with respect to the alternate hypothesis that the mean of the 2B-en HCRs is greater than the mean of 400M-HCRs). The results of this two-sample t-test are captured in Table 1.

As seen, for all the three categories of hateful, targeted, and aggressive speech, the strong T-values (14.48, 13.8 and 4.44) combined with high Cohen's-d (2.64, 2.47, 0.87) and low p-values (all $\ll 1e^{-4}$) strongly support the hypothesis that the file-wise HCR associated with the 2B-en dataset is higher than the file-wise HCR for the 400M dataset, thus adding further evidence to our claim of dataset degradation upon dataset scaling.

### 4.3 Connecting toxic alt-text and NSFW labels

The creators of LAION provide a 'Not Safe For Work' (NSFW) tag alongside each image to filter out undesirable content for the user. The score was computed using a custom-trained NSFW classifier built by the LAION team to detect harmful content in the dataset. As input, this classifier takes CLIP ViT L/14 embeddings, based on which it estimates a value between 0 (safe) and 1 (unsafe). As reported by the LAION team, on a manually annotated test set of 3000 samples, the accuracy of the classifier was 0.961. For filtering purposes, they recommend a safety threshold of 0.5, which allows excluding the most explicit images.

To establish a connection in terms of harmful content contained in either of the two modalities of the dataset, we measured the extent to which the images the LAION NSFW classifier (which uses only the images as input) flags as harmful are accompanied by alt-text that was tagged as harmful by *pysentimiento*. While the NSFW values provided for the LAION 400M dataset were discrete (`"UNLIKELY"`, `"UNSURE"` and `"NSFW"`), therefore precluding any values-based analysis, those provided for the LAION 2B-en dataset were continuous. We, therefore, focused on the LAION 2B-en

|  | NSFW >= 0.5 | | | NSFW <0.5 | | |
|---|---|---|---|---|---|---|
|  | hateful | targeted | aggressive | hateful | targeted | aggressive |
| **p ≤ 0.25** | 91.44% | 95.42% | 99.23% | 99.57% | 99.9% | 99.9% |
| **0.25 < p ≤ 0.5** | 2.31% | 1.86% | 0.54% | 0.24% | 0.06% | 0.10% |
| **0.5 < p ≤ 0.75** | 2.40% | 1.31% | 0.18% | 0.12% | 0.02% | 0.00% |
| **p < 0.75** | 3.85% | 1.40% | 0.04% | 0.12% | 0.01% | 0.00% |

Table 2: The percentage of values of hateful, aggressive, and targeted speech (as identified by *pysentimiento* in our sample of LAION 2B-en, divided based on the suggested NSFW threshold of 0.5. It can be seen that even the 'safe' subset (on the right) contains traces of toxic speech based on the alt-text.

dataset to carry out our analysis. To do so, we compared the probability of a given image being labeled as NSFW by the LAION safety classifier to the probabilities assigned for the three classes (hateful, targeted, and aggressive text) by *pysentimiento* for each image-text pair in our sample. We found a **slight correlation between 'hateful' and 'targeted' values and NSFW** ones, with a Pearson correlation coefficient of 0.227 and 0.215, respectively. The correlation with 'aggressive' values was much lower, at 0.076, with very low p-values (all $\ll 0.005$). This would indicate a slight but not substantial correlation between harmful content in images and that contained in their accompanying alt-text.

After filtering out the LAION 2B-en sample according to the suggested threshold of 0.5, some of the harmful content identified by *pysentimiento* does get removed, as shown in Table 2. However, there are still trace amounts of hateful, targeted, and aggressive content that remain, e.g., 0.24% of likely hateful content (with a probability larger than 0.5) and 0.03% of targeted content in the "safe for work" portion of the sample (right side of Table 2). This suggests that filtering out data only based on the NSFW image classifier alone does not filter out all hateful and targeted content. When carrying out qualitative observations of the portion of samples about and below the NSFW threshold, we also found examples of extremely toxic content in the alt-text (for instance, targeting women in terms of sexual violence), which was accompanied by inoffensive images. This suggests that using both text- and image-based safety filtering concurrently can help detect this kind of data and remove it before training.

We present further analyses regarding the relationship between the image and text safety filtering and provide more detailed comparisons between the datasets in Appendix B. All of our results, as well as the meta-dataset created as a result of our audit, are available in our repository. We hope that the meta-datasets we have generated will continue to be used by the community to guide further work.

## 5  Discussion and Recommendations

In this paper, we have systematically examined two datasets (LAION 400M and LAION-2B-en) and models trained on them. Contrary to current discourse in ML that treats scale as an unambiguously good attribute, our findings reveal that scale exacerbates hateful content. Datasets are not only fundamental to equitable, just, robust, and well-performing models, but also rigorous evaluation, audit, curation, and management of datasets is critical for advancing the field. Below, we present a set of observations, recommendations, and limitations of our study. We hope such discussions help the machine learning community, dataset creators/curators, and other stakeholders towards advancing not only data creation and curation but also the field as a whole in a manner that is transparent, rigorous, responsible, and accountable.

**Computational constraints:** As models and datasets get ever larger, machine learning becomes a field that is dominated by (and accessible to) a handful few within tech corporations and elite universities with unconstrained computational resources, crowding out those outside of it [1]. The presence of big tech-affiliated influential papers in machine learning, for example, shows an increase from 13% in 2008/09 to 47% in 2018/19 [6]. Assembling large-scale datasets requires relatively fewer resources, time, and effort than auditing, investigating, and "cleaning" them. Conversely, big tech corporations and large institutes with abundant computing power assemble these datasets. At the same time, most often, the thorough investigation and cleaning up is left to critical scholars with little resources. In this study, we have investigated as thoroughly as possible, given our relatively limited resources. We encountered various issues through manual investigation, such as poor data quality. For instance, an overwhelming number of images were screenshots. We could not perform a thorough analysis to determine a precise estimate of such

poor-quality data due to the cost of access to image APIs and the substantial computational resources required to download the datasets in their entirety to sift through them. Even when large datasets such as those that we have audited are open-sourced, getting the computational resources and tooling necessary for rigorous audit is a challenge. For instance, downloading LAION 2B-en requires 6.2TB of storage, with additional resources needed to run analyses such as running *pysentimiento* and the NSFW classifier. We encourage corporations and institutes to perform such audits. However, such self-audits will remain insufficient given the validity and credibility of self-audits are limited compared to those carried out by independent auditors [11]. Subsequently, we hope – perhaps through a coalition of the larger community, regulatory, and funding bodies – for the cultivation (through incentives) and creation of an ecology that allocates compute resources for independent auditors without access to institutional compute.

**The risks of extrapolation:** The $\delta_{CI}$ of $12.26\%$, calculated in Section 4.1 above, has important consequences on estimating the number of low-quality samples that either ought to be filtered out or at least re-investigated on account of having failed the text-quality mechanism that we have proposed. At a fixed $P_{threshold}$ of $0.5$, the estimated confidence interval (CI) for the number of failed-quality-test text samples in LAION-400M can be closer to 1.21 to 1.26 million samples. Suppose we use the upper bound of this CI as a benevolent estimate of HCR and extrapolate this to the LAION-2B-en dataset. It produces an upper-bound estimate of 7.1 million text-image pairs of hateful/aggressive/targeted content. However, as we have seen with our HCR analysis with actual 2B-en samples above in equation 3, the CI for this number is much higher, at 7.9 to 8.6 million samples. That is, our upper bound estimates based on the LAION-400M analysis turn out to be lower than the lower-bound estimate obtained by looking at LAION-2B-en samples by a margin of 0.865 million images (See Figure A1 in the Appendix). This massive margin illustrates that extrapolation using confidence intervals, especially on datasets with underlying graph structure and rich inter-node correlations such as the Common Crawl where the sample-level i.i.d. (Independent and Identically Distributed) assumptions may be invalidated, is imprudent and may lead to under-estimation of hateful/aggressive/targeted content.

**The dangers of thresholding:** As we show in Section 4.3, even a conservative NSFW threshold of 0.5 still keeps a certain amount of toxic and hateful content in alt-text, on top of the false negatives it may have for images. Given that for some LAION releases, discrete safety tags were provided based on pre-defined thresholds like this one, there will be consequences on downstream data quality (given that data users may filter the dataset based on the discrete values). Also, above and beyond false negatives, any predefined threshold assumes that a set amount of NSFW content is universally acceptable regardless of downstream data usage, which should be a decision for the data users. Therefore, we advocate that the raw values of *multiple types* of hateful content detection approaches (based on text and images) are provided with datasets, alongside evaluations of what kind of content different thresholds filter out. The final thresholding values would then be established by dataset users, who can make informed decisions based on the provided information – for instance, they may filter out more NSFW and toxic content if training a text-to-image model that illustrates children's stories, as opposed to one that generates stock photos. This would require more fine-grained evaluation than the one currently carried out for content filters, which does not analyze filtered content and uses accuracy as the sole metric for measuring success.

**Appropriate and consistent metrics:** Even though many of these datasets are created to train semantic search systems and image generation models that supposedly "democratize art-creation" for the general public (where a significant proportion are people of diverse gender, ethnicity, and race) the metrics used to check if progress is indeed being made by dataset scaling rarely reflects that diversity. While specific analyses are being made regarding the risk of biases and harm in ethics and safety subsections of reports and papers accompanying datasets, the metrics that supposedly measure these harms are rarely incorporated as part of the model check-pointing process. For instance, in the ALIGN paper [33], the dataset scaling ablation study focused only on two metrics: the MS-COCO zero-shot retrieval accuracy rates (I2T-$R@1$ and T2I-$R@1$) and the ImageNet K-Nearest-neighbor (KNN) $R@1$ rates. In the BASIC model paper ablation study [52], the authors gauge the impact of increasing the dataset size from 1.7B to 6B by comparing the ImageNet-1k zero-shot top-1 accuracy. Finally, in the LAION-5B paper [62], the authors use the zero-shot top-1 classification accuracy metric, once again on the ImageNet family of datasets (with the distribution shift variants) and a bespoke VTAB+ benchmark spanning 35 classification tasks covering different computer vision datasets. This means that it is difficult for users to meaningfully compare the metrics and performance on any of these datasets to each other without re-running analyses. Using standardized, meaningful metrics for measuring progress is vital to ensure that we are indeed measuring what we set out to measure and that results are comparable and reproducible.

**Exploiting multimodality in toxic content detection:** While there has been some work in multimodal hate speech in specific contexts such as memes [39, 69] as well as on topics such as cyber-bullying [37], we were not able to find existing multimodal approaches that were generic enough to be usable for creating multimodal datasets such as LAION. Our study intended to connect the two modality-specific approaches described in Section 4.3. However, there is much work to be done in connecting the type of content that is often the focus of image-based filtering (for instance, violence and sexually explicit content) as well as the focus of text-based filtering (for example, hate speech and aggression). It is crucial to understand the links between images and text better and to use them both when filtering datasets. Finally, no kind of filtering is, in itself, a bulletproof solution against harmful content. Without careful contextual analysis, filtering mechanisms can censor and erase marginalized languages, identities, and experiences [16].

**Limitations of pysentimiento:** The analysis of hate content and scale in this paper was performed through the *pysentimiento* library. This library is known to have high predictive accuracy and works in multiple languages [51]. However, as far as we know, there has been no research on the variability of *pysentimiento* regarding linguistic variants of English or specific topics. To continue rigorous audits and improve multimodal-toxicity detection models, we encourage future work to use these various NLP models to investigate the impact of scale on hateful content and confirm if our finding of hate-scaling holds upon using different NLP models. This, for example, includes further investigation to establish if the alternative hate detection options, including open-source models (e.g. [71, 24, 27, 13]) produce different or complementary results to ours.

**Legal and policy implications:** The multimodal datasets we audited form a crucial backbone for modern machine learning systems, including generative models. These models are not purely intellectual exercises but are integrated into society, directly or indirectly impacting actual people. Subsequently, legal issues arise from multiple angles, including consent and rights of individuals in datasets, what should be in datasets and how they should be evaluated and maintained, and mechanisms for responsibility and accountability for problematic content in the dataset as well as the downstream effect on models trained on it. Closing access to datasets used for popular and impactful models and active obfuscation of information around these datasets present a significant obstacle to developing appropriate regulatory guidelines and guardrails. This audit study showed extensive evidence of the exacerbation of hateful content correlated with scale. We hope this work serves as an initial document for legal and policy experts alike that both demystifies multimodal datasets and illustrates the negative implications of scale.

# 6 Conclusion

We have conducted a dataset audit of two vision-language multimodal datasets, LAION-400M and LAION 2B-en, and presented evidence of hateful, aggressive, and targeted content, which was exacerbated by dataset size. This type of analysis was not previously carried out by the dataset creators, who focused on analyzing the characteristics of the images and not their accompanying alt-texts – our analysis has found that these two types of analyses are correlated but complementary. We conclude with some final remarks: we cannot stress enough the importance of open-source in audit endeavors such as ours since any quantitative and qualitative dataset exploration hinges upon access to the artifacts themselves. Providing access to datasets and models is essential to a healthy, thriving research community. However, we are saddened that an increasing number of machine learning organizations fail to do so.

Today's state-of-the-art vision-language multimodal models are trained on datasets like the ones we examined in the present article. These models are currently being deployed in real-world recommendation systems, information-retrieval systems, semantic search systems, and image captioning systems, despite them exhibiting biases and stereotypes [5, 42] that are neither well understood nor documented. Given that such failures can result in dire consequences on real people, often those at the margins of society, we implore the research community as well as those developing and deploying these systems to carry out due diligence with rigorous audits of these models and training datasets and take necessary actions, including refraining from use in high-stake scenarios.

**Acknowledgements:** Vishnu Naresh Boddeti was supported by the National Science Foundation under Grant No. (#2147116).

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

## Supplementary Materials

## A   The risks of extrapolation

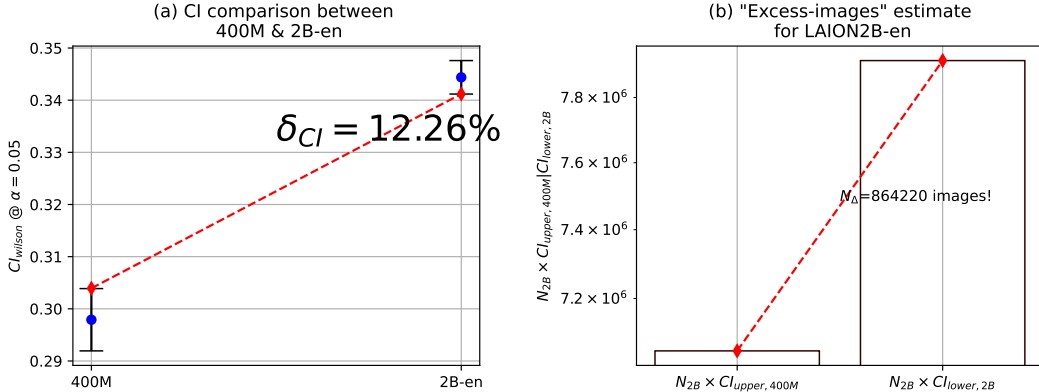

Figure A1: Binomial proportion confidence interval (CI) analysis to establish the extent of HCR underestimation upon using LAION400M statistics.

## B   NSFW Analysis

|          | NSFW >= 0.5 | | | NSFW <0.5 | | |
|----------|---------|----------|------------|---------|----------|------------|
|          | **hateful** | **targeted** | **aggressive** | **hateful** | **targeted** | **aggressive** |
| **mean** | 0.749   | 0.356    | 0.104      | 0.021   | 0.013    | 0.011      |
| **min**  | 0.501   | 0.008    | 0.013      | 0.00    | 0.013    | 0.005      |
| **max**  | 0.988   | 0.961    | 0.894      | 1.00    | 0.499    | 0.513      |

Table A1: The mean, minimum, and maximum values for hateful, targeted, and aggressive content in our sample of LAION 2B-En, are divided according to the suggested NSFW threshold of 0.5.

## C   The origins of the dataset scaling laws: A cartoon sketch emerges

While attempting to unearth what this specific dataset scaling law was that the practitioners were so inspired by, we repeatedly encountered a certain cartoon sketch 'power-law' plot referred to in both personal exchanges as well as in surveys such as [72]. As it turns out, this cartoon sketch power-plot first appeared as `Figure 6` in *"Deep learning scaling is predictable, empirically"* [29], a work that emerged out of Baidu research in 2017. The authors that first presented this plot posit that the generalization error associated with an ML model exhibits a three-phase behavior about its training dataset size. In the first phase, they state maps to the 'small data region,' where *"models will struggle to learn from a small number of training samples"* resulting in high generalization errors. The second phase (or the middle portion of learning curves), they claim, is the 'power-law region,' where the generalization error monotonically decreases with training dataset size (linear with application-specific slopes when plotted on a log-log scale). This phase stretches till we hit the point of the 'glass-ceiling' or 'unbreachable error floor' on account of factors such as model mismatch and mislabeled data (constituting the third phase). This, of course, has been further supplanted by the likes of the Chinchilla scaling laws (20 tokens per model parameter) [30] in the specialized context of LLMs.

## D   Blackbox non-reproducible empirical results

As for the black box non-reproducible empirical results that validated the dataset-scaling mandate and championed the *scale-beats-noise* narrative, we refer to the ALIGN paper [33] that emerged in 2021. In

the abstract section of this paper, we first encounter the following claim: *"We show that the scale of our corpus can make up for its noise and leads to state-of-the-art representations even with such a simple learning scheme"*. The demonstration of this claim appears later in ``Section 6.2. Pre-training Datasets'' where the authors state that *"To understand better how data size scaling wins over the increased noise, we further randomly sample 3M, 6M, and 12M ALIGN training data and compare them with the cleaned CC-3M data on B7+BERT-base model. Table 10 (sic) shows that while the ALIGN data performs much worse than CC data with the same size (3M), the model quality trained on 6M and 12M ALIGN data rapidly catches up. Despite being noisy, ALIGN data outperforms Conceptual Captions **with only 4x size**."* We note that these experiments (or similar ones) have not been replicated elsewhere to check if these scaling ratios presented *ipse dixit* in these contexts indeed hold at all.

## E    The tactical template: Fuzzy main section meets non-existent appendices

What unites the marquee projects of Dall-E, Parti, and Imagen is the near-same tactical template deployed when it comes to (non)declaring the training dataset information. The template runs something like this:

**Step-1:** Allocate a small nondescript subsection of the main section of the paper covering only the bare minimum details about the number of samples in the training dataset with cross-references to other similar black box datasets such as JFT. This coincidentally happens to be Section 4.1 in both Parti and Imagen papers (See Fig A2).

**Step-2:** Declare that somewhere in the succeeding sections titled on the lines of broader impacts or societal impacts are details about the 'potentially problematic' aspects of the dataset and the downstream risks while patronizingly citing previously published audit papers (such as [7] that have done the grunt work of exposing the gory details of such datasets. This happens to be Section 8 - Broader impacts in Parti and Section 6 for the Imagen model.

**Step-3:** Setting the reader up for a non-existent Appendix section that is not part of the main paper and does not contain any details about how the dataset is constructed and where the data is sourced from while noting the fact that it's not mandatory for the reviewers to even glance at the Appendix section in peer-reviewed avenues of publishing.

It is in this backdrop we observe that the authors of the BASIC model paper have not even addressed model safety and dataset auditing issues despite having trained their model on the largest image-text dataset ever assembled and presented a full-length 47-page paper with three revisions on ArXiv (See https://arxiv.org/abs/2111.10050).

# Scaling Autoregressive Models for Content-Rich Text-to-Image Generation

## 4.1 Training Datasets

We train on a combination of image-text datasets for all Parti models. The data includes the publicly available LAION-400M dataset [43]; FIT400M, a filtered subset of the full 1.8 billion examples used to train the ALIGN model [9]; JFT-4B dataset [44], which has images with text annotation labels. For textual descriptions of JFT, we randomly switch between the original labels as text (concatenated if an image has multiple labels) or machine-generated captions from a SimVLM model [45]. We discuss the limitations of the data in Section 8. For all image inputs, we follow the DALL-E dVAE input processing (Section A.2. Training in [2]) for image tokenizer training and the DALL-E Transformer input processing (Section B.2. Training in [2]) for encoder-decoder training.

## 8 Broader Impacts

**Bias and safety.** Text-to-image generation models like GLIDE, DALL-E 2, Imagen, Make-a-Scene, CogView and Parti are all trained on large, often noisy, image-text datasets that are known to contain biases regarding people of different backgrounds. This is particularly highlighted in Birhane et al's [100] analysis of the LAION-400M dataset [43]: their study of the dataset surfaced many problems with respect to stereotyping, pornography, violence and more. Other biases include stereotypical representations of people described as lawyers, flight attendants, homemakers, and so on. Models trained on such data without mitigation strategies thus risk reflecting and scaling up the underlying problems. Our primary training data is selected and highly filtered to minimize the presence of NSFW content; however, we incorporated LAION-400M during finetuning with classifier-free guidance – this improved model performance but also led to generation of NSFW images in some contexts. Other biases include those introduced by the use of examples that primarily have English texts and may be biased to certain areas of the world. In informal testing, we have noticed, for example, that prompts mentioning wedding clothes seem to produce images biased towards stereotypically female and Western attire.

# Photorealistic Text-to-Image Diffusion Models with Deep Language Understanding

## 4.1 Training details

We train on a combination of internal datasets, with ≈ 460M image-text pairs, and the publicly available Laion dataset [61], with ≈ 400M image-text pairs. There are limitations in our training data, and we refer the reader to Section 6 for details. See Appendix F for more implementation details.

## 6 Conclusions, Limitations and Societal Impact

Imagen's training data was drawn from several pre-existing datasets of image and English alt-text pairs. A subset of this data was filtered to removed noise and undesirable content, such as pornographic imagery and toxic language. However, a recent audit of one of our data sources, LAION-400M [61], uncovered a wide range of inappropriate content including pornographic imagery, racist slurs, and harmful social stereotypes [4]. This finding informs our assessment that Imagen is not suitable for public use at this time and also demonstrates the value of rigorous dataset audits and comprehensive dataset documentation (e.g. [23, 45]) in informing consequent decisions about the model's appropriate and safe use. Imagen also relies on text encoders trained on uncurated web-scale data, and thus inherits the social biases and limitations of large language models [5, 3, 50].

Figure A2: The Google template used to (non)declare the training dataset information along with paper screenshots

