# OpenReview forum: "Into the LAION’s Den: Investigating Hate in Multimodal Datasets"
_NeurIPS.cc/2023/Track/Datasets_and_Benchmarks — NeurIPS 2023 Datasets and Benchmarks Poster_

### Official Review · Reviewer_gEqv · 2023-07-15
**An evaluation of data scaling and it consequences regarding hateful content**

**Rating:** 4
**Confidence:** 3
**Correctness:** 1. Reference 53 and 54 are the same.
…
**Clarity:** 1. What is "160 (= 32+128)" in line 145?

**Strengths:**

1. The paper touches base on a hot topic - the impact/limitation of data scaling on model performance and society.
2. It analyses multimodal datasets.

**Additional Feedback:**

N/A

**Documentation:**

yes

**Limitations:**

1. The comparison does not seem quite valid to me as the data size for the two datasets is significantly different. A larger dataset is likely to contain more noisy data. Instead, the data is better sampled proportionally.
2. Using only one measurement is not sufficient. The authors mention in section 5 that pysentimiento lacks research on its variability in terms of linguistic variants of English or specific topics. Hence it is questionable to interpret the reliability of results if no other models/metrics are used. The selection of pysentimiento is not very much justified. Is it better than other models/APIs like perspective? For choosing pysentimiento why did you mention misogyny and racism which is neither discussed in the rest of the paper nor used for the analysis of the LAION datasets?

**Opportunities For Improvement:**

I would also suggest including/discussing Bloom in the paper. While it is not multimodal, the initiative is similar to what the authors advocate, open access and good documentation; however, its performance is not so great compared to other models. It would be worth discussing this aspect.

**Relation To Prior Work:**

yes

**Summary And Contributions:**

This paper's argument is centred on the narrative that scaling shouldn't be the priority when developing data. The authors try to back up this idea by showing the amount of abusive text increases as the dataset scales up. However, the authors did not show how a large amount of hate speech in the dataset would hurt model performance. A better way to show this is to relate data size with data quality and model performance. For instance, analyse whether removing the problematic data leads to performance improvement. If you don't show this, I believe it will not 'convince' big corporates to change their strategies.

Furthermore, another important issue here, as the authors point out, is annotation is expensive which is also why scaling is mostly adopted. It would be better if the authors show what the effect of annotation is on scaling vs. model performance.

---

> ### Author Response · Authors · 2023-08-20
>
> Reviewer comment: “...the authors did not show how a large amount of hate speech in the dataset would hurt model performance. A better way to show this is to relate data size with data quality and model performance.”
> Response: We thank the reviewer and acknowledge this is an important point. Our longer version of this project extends into followup model audit and examines the impact of dataset size on model performance. However, due to the limited space of a NeurIPS D& Bpaper format, we were not able to include both dataset and model audits into one paper. As a result, model audit (which we have developed as a related but separate work), falls outside the scope of this paper.
>
> R comment: “... annotation is expensive which is also why scaling is mostly adopted. It would be better if the authors show what the effect of annotation is on scaling vs. model performance."
> Response: We thank the reviewer for this suggestion and agree that it is another interesting topic of study. However, this again is outside the scope of the current paper.
>
> R comment: "I would also suggest including/discussing Bloom in the paper."
> Response: We thank the reviewer for bringing this up. We do refer to the BLOOM training corpus, ROOTS, in our paper. We will furthermore, incorporate relevant aspects of the BLOOM model paper in the next and final iteration.
>
> R comment: "The comparison does not seem quite valid to me as the data size for the two datasets is significantly different. A larger dataset is likely to contain more noisy data. Instead, the data is better sampled proportionally."
> Response: We thank the reviewer for this critique. Firstly, we would like to better understand the intellectual basis behind the ansatz proposed here: ”A larger dataset is likely to contain more noisy data”. As you might fathom, this is a rather strong claim and we were curious if this was a personal belief or if you could point to certain references. Secondly, one of our concerns that we have alluded to in Section 2.1 is the prevalent practice of using the term ‘noise’ as a tactically benign neologism. While ‘noise’ may make sense in the case of supervised learning with categorical labels (mislabeling ~ noise), in the context of alt-text based multimodal images however, it cheapens and confuses the exact nature of hate that is fuelling the textual deviations from a normatively acceptable ‘ground truth’ explanation of the contents of an image. For example, in the datasets we audited, we encountered an image from “https://see.xxx/mt/sL/1994633.jpg” that had an alt-text description: "Hirsute wet crack of this gorilla lady is so nasty that dont crave to fuck that". We strongly feel uncomfortable with the categorization of textual description like this as being merely ‘noisy’. With regards to the “sampled proportionally” critique, we would like to reemphasize that: The two datasets are not monoliths. In fact, they are both collections of parquet shards (32 for LAION-400M and 128 for LAION-2B). While the same code (that sampled 0.1 million random rows from each parquet file and passed them through the NLP model) was used to evaluate the metrics at the shard-level, the data analysed does in fact scale (if not precisely proportionately) as the number of parquet shards is 1:4 between LAION400M and LAION2B. This is also precisely why we have presented the Fused swarm-box-violinplot that captures the *file-wise* HCR metrics in Figure 2.
>
> R comment:"The selection of pysentimiento is not very much justified."
> Response: We wholly agree with the reviewer that using multiple metrics can be a better indication of multiple aspects of models and datasets. In fact, we used pysentimiento because it outputs multiple scores (`hateful′,′targeted′,′aggressive`) when queried, whereas many other tools and libraries only output a single score. Expanding our work with different other tools could be interesting future work.
>
> R comment: "Reference 53 and 54 are the same. Line 331: be provided -> are provided? Line 345: Study -> study."
> Response: We thank the reviewer for flagging such errors. We have fixed references 53/54 and the other two points that were raised.
>
> R comment: "Line 361: please remove "extensive" as I don't think you did an extensive analysis."
> Response: We thank the reviewer for raising this criticism of our work. To the best of our knowledge, our paper represents the most in-depth analysis of hateful content in the LAION datasets, as well as the only analysis that connects both the “Not Safe For Work” content that is contained in the images and the hateful content in their accompanying captions. We are, therefore, convinced that our analysis and the code and data that we are sharing along with it are important artefacts that can benefit the community.
>
> R comment: "What is "160 (= 32+128)" in line 145?"
> Response: This refers to the number of files that the LAION datasets were separated into, which we used for our analysis. We will make this more clear in the text.

---

### Official Review · Reviewer_9brB · 2023-07-20
**Good paper that can be Great with a Couple Edits**

**Rating:** 8
**Confidence:** 5

**Strengths:**

The perspective of this paper is refreshing. There is an amazing need for explainability and auditing tools in the NLP/LLM space due to the massive explosion of Generative AI and the ever growing datasets that are used to build it. I also love that they want for the code resulting from this work to be open source. It is very important in these spaces to build upon prior work.

**Additional Feedback:**

NA

**Clarity:**

The paper has some minor grammatical errors but nothing major. For example, in the conclusion a sentence reads "We conclude with some final observations: first, we cannot stress the importance of open-source in audit endeavors such as ours, since any kind of quantitative and qualitative dataset exploration hinges upon access to the artifacts themselves." (pg. 9) When I am sure they mean "...first, we cannot stress [enough] the importance." This happens a few times in the paper, but not enough to make it difficult to read or understand.

**Correctness:**

I am not fully confident that the proba and pysentimiento packages are completely reliable detectors of hate speech for the reasons I state in the Limitations section of this review. That said, the actual analysis shown in the code is in line with what the authors report in the paper.

**Documentation:**

Yes

**Ethics:**

I see the authors sited their use of pysentiento but want to make sure they properly site the use of probas which is used in their code to calculate the probability hate variable in the code. This code is an open sourced offering that assist with identifying bad labels and should be acknowledged as well in the text.

**Limitations:**

The authors do acknowledge that multi-modal models would need auditing on both linguistic and visual components of their dataset for proper auditing, something they do not do in the paper. The paper focuses on simply the linguistic portion. They also discuss the limitation of access to datasets of most major LLMs and how this can affect the ability to audit these models effectively. Unfortunately, they miss an important limitation of their analysis- their dependence on packages such as pysentimiento and proba that may incorrectly label certain things as hate speech or having a negative sentiment that may not be. One such example of this happening within the paper is the sentence the authors use in their auditing methodology paragraph: ‘slave punished by angry blond mistress’ that is rated with a .96 hate speech score- no doubt because of the inclusion of the word 'slave'. The problem is, what is actually hateful about that sentence? Was this sentence used when discussing an historical fact? If so, slaves WERE punished by their white slave owners that may have been blond and, yes, may have been angry. Is the problem the word 'mistress'? It is not clear what makes a sentence such as that hateful, and, as such, there can be other sentences labeled 'hateful' that are not hateful at all though they may make one uncomfortable to read. Slavery is a fact in this country and any model that tries to flag references to it may end up causing more harm than good.

**Opportunities For Improvement:**

The work does not reference the amazing dive into the problems with gathering data from the internet using services like Common Crawl by the authors Dr. Emily Bender and Dr. Timnit Gebru in the seminal work "On the Dangers of Stochastic Parrots: Can Language Models Be Too Big?". It would be to the benefit of the authors to include this very important work and to add it to their references as it assist with many of their claims in the paper and can help them strengthen their concluding arguments about the importance of auditing LLMs as a social good.

In that same vein, the authors should be more clear with what they see as harms and what their auditing mechanism can do to solve said harms. I find the authors hint at the problem with what is designated as hateful speech without actually telling the reader how this hateful speech can affect their daily lives or help fight against model degradation/improve model performance, etc. There are many perspectives they can take, but whichever they chose to really focus on should be one that truly shows the reader why they should care and use these auditing tools in the future.

**Relation To Prior Work:**

While there is reference to prior related work done in this space within the Overview section, it can be useful of the authors to add a sentence or two really fleshing out how this work builds on the groundwork already performed. How is this paper positioned between the camps pushing for scaling as a means to gaining better model performance and the camps calling for additional auditing of data sources and transparency of model decision making? How does it reconcile or reconsider the juxtaposition of these two ideas?

**Summary And Contributions:**

This paper provides an auditing tool for the two LAION datasets that feed multiple Generative AI models. The authors include supplementary code that shows how they are able to detect hateful content in these large datasets and show that the probability rate of hate content within these labels are larger in larger datasets perhaps signaling that hate speech is scaling right along with dataset size, compute and model parameters.

---

> ### Author Response · Authors · 2023-08-20
>
> Reviewer comment: "The work does not reference the amazing dive into the problems with gathering data from the internet using services like Common Crawl by the authors Dr. Emily Bender and Dr. Timnit Gebru in the seminal work "On the Dangers of Stochastic Parrots: Can Language Models Be Too Big?"
> Response: We agree with the reviewer that this is an important reference and we have incorporated it into our paper.
>
> Reviewer comment: "I find the authors hint at the problem with what is designated as hateful speech without actually telling the reader how this hateful speech can affect their daily lives or help fight against model degradation/improve model performance, etc."
> Response: We agree with the reviewer and will include an explainer on what hateful speech is and the concrete impacts on daily lives.
>
> Reviewer comment: "The authors do acknowledge that multi-modal models would need auditing on both linguistic and visual components of their dataset for proper auditing, something they do not do in the paper. The paper focuses on simply the linguistic portion."
> Response: We agree with the reviewer that multimodal models and datasets can be audited for various issues from various angles. However, given the vastness of the datasets we are auditing and the limited format of the NeurIPS D&B track paper template, the tradeoff we had to make was an in depth analysis on a specific issue in a focused manner or cover multiple issues but in a relatively less thorough manner. We believe an in depth and thorough audit on a specific aspect of the dataset (the linguistic portion in this case) is the most informative, hence, our focus on it.
>
> Reviewer comment: "They also discuss the limitation of access to datasets of most major LLMs and how this can affect the ability to audit these models effectively. Unfortunately, they miss an important limitation of their analysis- their dependence on packages such as pysentimiento and proba that may incorrectly label certain things as hate speech or having a negative sentiment that may not be."
> Response: We acknowledge that tools such as pysentimento and proba are not entirely accurate. In fact, no such tools can ever be accurate due to the ambiguous and messy nature of human language and hateful content; they defy simple automation and accurate classification. Making the idea of a perfect tool impossible. To this end, we will add a disclaimer emphasising the limitations of the tools we are using.
>
> Reviewer comment: "I am not fully confident that the proba and pysentimiento packages are completely reliable detectors of hate speech for the reasons I state in the Limitations section of this review. That said, the actual analysis shown in the code is in line with what the authors report in the paper."
> Response: We agree with this and have addressed it in the above comment.
>
> Reviewer comment: "The paper has some minor grammatical errors but nothing major."
> Response: We appreciate the feedback on grammatical errors contained in our paper and we have corrected those identified by the reviewer. We generally aspire to make the camera ready as clean and correct as possible.
>
> Reviewer comment: "While there is reference to prior related work done in this space within the Overview section, it can be useful of the authors to add a sentence or two really fleshing out how this work builds on the groundwork already performed. How is this paper positioned between the camps pushing for scaling as a means to gaining better model performance and the camps calling for additional auditing of data sources and transparency of model decision making? How does it reconcile or reconsider the juxtaposition of these two ideas?"
> Response: We agree with the reviewer that these are important questions. We hope our paper contributes (although does not answer them directly) by bringing clarity around scale and its impact on hateful content. As stated in the paper, we particularly wanted to investigate if claims such as “scale balances out bias” in fact stand up to scrutiny. As a result, our analysis is focused on dataset contents, not on models. Furthermore, our goal is not necessarily to help model creators produce more accurate or performant systems, but those that are less likely to generate hateful content. It is logical that removing this content from their training data contributes towards this goal (since models “see” less hateful content during training, they are less likely to produce it during inference).
>
> Reviewer comment: "I see the authors sited their use of pysentiento but want to make sure they properly site the use of probas which is used in their code to calculate the probability hate variable in the code. This code is an open sourced offering that assist with identifying bad labels and should be acknowledged as well in the text."
> Response: We thank the reviewer for picking up on this unintentional omission. We will cite proba in the next iteration.

---

### Official Review · Reviewer_ToJe · 2023-07-21
**The Presence of Hate in Multimodal Datasets**

**Rating:** 7
**Confidence:** 3
**Correctness:** It looks perfectly accurate and correct.
**Clarity:** The paper is exceptionally well-written.

**Strengths:**

The topic is timely, the paper is extremely well-written, motivated, and explained. The analyses, are sound and they mostly show clear results. Large datasets are also processed in those analyses.

Figure 1 is very impactful.

**Additional Feedback:**

%

**Documentation:**

The code is provided and instructions for how to download the data.

**Ethics:**

No.

**Limitations:**

Yes, I cannot think of any other key limitations that are missed.

**Opportunities For Improvement:**

My main comment is about hateful content being only inferred from the alt-text and not from images. However, there is the previous work on images, which also tags them, allowing for a comparison with this work.

Another smaller issue that I have is claiming that for sure scale is the only / main reason for the increase in hate speech. Could it be that a lot of hateful was added while scaling from LAION-400M to LAION-2B, but that this would not necessarily happen for some other models if the data are chosen more carefully?

There could be an explanation on what exactly the three measures capture: hateful, targeted, and aggressive?

Line 319 is missing Figure number.



**Relation To Prior Work:**

OK.

**Summary And Contributions:**

In this work, two multimodal datasets that are similar but of different scale (LAION-400M and LAION-2B) are compared in terms of presence of hateful speech in the alt-text accompanying images. The result is that for various measures of the hateful text, the scores increase in the larger model.

---

> ### Author Response · Authors · 2023-08-20
>
> Reviewer comment: "My main comment is about hateful content being only inferred from the alt-text and not from images. However, there is the previous work on images, which also tags them, allowing for a comparison with this work."
> Response: We wholeheartedly agree with the sentiment conveyed here. Auditing multimodal datasets can be carried out along different dimensions addressing each data-modality or tackling a combinatorial tuple of many such modalities. We have, in fact, dedicated the entirety of the first paragraph In Section 3.1 titled ‘Audit methodology’ (Lines 118 TO 125)  addressing this issue:
>  “One of the challenges of auditing multimodal datasets is that harm and biases can be present in either  modality. Most of the audits of multi-modal datasets thus far have investigated the images contained within them, using techniques ranging from image content analysis [55, 5] to image source analysis (by analyzing the URL field), as well cross-modal analyses between images and text (i.e. looking for discordance between an image and its alt-text description). These analyses have indicated poor data quality for instance, recent work found that nearly 30% in the LAION-2B-en dataset are duplicates (700 million image-text pairs) [62] . Other work has shown the presence of explicit images, pornography, 125 malign stereotypes and other extremely problematic content in the LAION-400M dataset [5]”.
> We’d also like to take this opportunity to highlight one prickly issue that emerges with auditing the *content* of images in datasets such as LAION . Given that there is a vast plethora of vividly pornographic images in the dataset, many with alt-text content such as ‘nubile’ and ’barely legal’, it is impossible to ascertain if the ethicist is unwittingly grappling with child-pornographic data under the aegis of academic inquiry. So, if we are to analyse the content of the image, this will necessitate downloading the image (or temporarily caching it) before batching the images and feeding the image tensor through the pretrained models under critique, which inturn means that the researcher is undertaking legal risks with stark ramifications.Hence, we harnessed the pre-generated NSFW values as a proxy and produce the results captured in Table-2 of the paper.
>
> Reviewer comment: "Another smaller issue that I have is claiming that for sure scale is the only / main reason for the increase in hate speech. Could it be that a lot of hateful was added while scaling from LAION-400M to LAION-2B, but that this would not necessarily happen for some other models if the data are chosen more carefully?"
> Response: We agree with the reviewer that there are multiple potentially confounding factors leading to an increase in hate speech and we acknowledge that scale is not the only factor. For example, the CLIP-threshold value that seems to have been reduced from 0.3 to 0.28 in a rather ad hoc fashion. We address this in lines 183 to 188 on Page 5 where we state: “Given that both LAION-400M and LAION-2B-en are extracted from the Common Crawl dataset, we hypothesize that during the race to expand the dataset to 5 billion samples, the dataset scraping module might have sampled from the lower-quality sub-parts of the Common Crawl at a rate worse than that during the LAION-400M curation process. We also note that the CLIP-filtering threshold seems to have been relaxed from 0.3 (for LAION-400M) to 0.28 (for LAION-2B-en), which could be another explanatory factor”. In this paper, we study the effect of scale, and find increased levels of hate as scale increases from LAION-400M and LAION-2B. We posit that, to a large degree, hate content in the training data leads to increased hate in the model. Therefore, more careful chosen data is a plausible solution to mitigate the problem. However, this may not be practically viable for extremely large datasets.
>
> Reviewer comment: "There could be an explanation on what exactly the three measures capture: hateful, targeted, and aggressive?"
> Response: We agree such an explanation brings further clarity to our paper and plan to incorporate it.
>
> Reviewer comment: "Line 319 is missing Figure number."
> Response: This and other grammatical errors and missing/incomplete information has been corrected.

---

> > ### Comment · Reviewer_ToJe · 2023-08-29
> >
> > The authors have addressed most of my comments (but not yet the one on adding explanations on what exactly the three measures capture: hateful, targeted, and aggressive). I will keep the rating as is.

---

> > > ### Author Response · Authors · 2023-08-29
> > > **Description of Three Measures in Pysentimiento**
> > >
> > > The pysentimiento model we used for detecting the three measures was trained on the dataset released by [1].
> > >
> > > We will add a paragraph describing the three measures in the paper: hateful, targeted, and aggressive, and provide an example sentence for each measure, such as those provided in the paper [1].
> > >
> > > To elaborate, the training dataset contains the following labels.
> > > - Hateful: A binary value indicating if hate-speech occurs against one of the given targets (women or immigrants).
> > > - Targeted - If hate speech occurs, a binary value indicates if the target is a generic group of people or a specific individual.
> > > - Aggressiveness - If hate speech occurs, a binary value indicates whether the tweeter is aggressive.
> > >
> > > We hope the above description addresses your concern, and please let us know if there are any further outstanding concerns.
> > >
> > > [1] [SemEval-2019 Task 5: Multilingual Detection of Hate Speech Against Immigrants and Women in Twitter](https://aclanthology.org/S19-2007) (Basile et al., SemEval 2019)

---

### Author Response · Authors · 2023-08-20

We thank the reviewers for such thorough reviews and extensive feedback. We particularly appreciate the reviewers' discussions of the paper’s strengths – i.e. its timeliness, impact, relevance, significance, clarity, and open-sourcing of code and data. We present responses to each reviewer's thoughtful questions, concerns, and identified opportunities for improvements below (under each reviewer comment), which we also incorporate into the camera ready version of our paper.

---

### Decision · Program_Chairs · 2023-09-22

**Decision:**

Accept (Poster)

**Comment:**

Because reviews were mixed and only 3 of the ideal 6 were provided, I read the paper as well. I had a very similar impression as Reviewer ToJe, and have some (strong) disagreements with statements made by Reviewer gEqv. To summarize the views of the 3 reviewers + me:

Strengths:
- Very timely topic.
- In an important domain.
- At times writing is passionate, but written well. 🔥
- Sound analysis.

Opportunities For Improvement:
- More explanation of why hate speech is an issue in this domain.
- Using only one main system (pysentimiento) to derive general conclusions is not ideal. Either that choice should be better justified, or more than one system should be used.
- Explanation on what exactly the three measures capture: hateful, targeted, and aggressive.
- Make it more clear throughout that the packages are not completely reliable. In particular, I noticed several places where the authors write results about "hate"/"harm" where I think they mean *predicted* hate/harm.

Limitations: Well written. Would have appreciated them being referred to earlier, such as noting that discussion on issues with pysentimiento are available in the Limitations section.

Correctness: Mostly correct. Not mentioned by other reviewers, but did you really intend for HCR to be multiplied by 100? The numbers provided appear to scale between 0 and 1, whereas I had thought you were scaling between 0 and 100(%). I had the sense that you were trying to convert probabilities to percentages, but then didn't end up doing it. You also describe it as a "rate" and "ratio" but I think I would call it an "average" (it's divided by the number of samples, giving you something like "average hate content"). I'd imagine a ratio to have the denominator as #samples - numerator.

Clarity: Good

Relation to Prior Work: Suggestions from reviewers of additional things to include.

Documentation: Reasonable

Additional comments:
- " dataset degradation " -- perhaps be more specific. More hateful?
- " precluding any kind of values-based analysis " -- Confusing. You mean any kind of *real* or *continuous*-valued analysis? The values given are ordinal. Not to mention the additional confusion with "value" often meaning peoples' values in this context.

Some typos (not all):
- 129: "such the ones"
- 139: "and scaleand to what harmful" (Also, I think by "what" you mean "whether"?)
- 168: "the amount of problematic"
- 169: "We also and"
- iid: You define in the second usage, not the first; flip.